# The evaluation of indoxyl sulfate in the general population in Kanegasaki Iwate: A cross-sectional study (KANEGASAKI study)

Mizuki Hisano[1], Takaya Abe[1]*, Kie Sekiguchi[1], Eri Hakozaki[2], Fumiaki Takahashi[3], Kozo Tanno[4], Toshikazu Abe[2], Kenichi Takeuchi[5], Toru Maruyama[6], Ryo Takata[1], Wataru Obara[1]

1 Department of Urology, Iwate Medical University, Yahaba, Japan, 2 The Kanegasaki Town Health and Welfare Center, Kanegasaki, Japan, 3 Department of Information Science, Iwate Medical University, Yahaba, Japan, 4 Department of Hygiene and Preventive Medicine, Iwate Medical University, Yahaba, Japan, 5 Iwate Health Service Association, Morioka, Japan, 6 Department of Pharmacology, Kumamoto University, Kumamoto, Japan

* takayabe@iwate-med.ac.jp

## Abstract

Chronic kidney disease (CKD) significantly impacts global health. Renal function is evaluated using urinary albumin–creatinine ratio (U-Alb/Cr ratio) and estimated glomerular filtration rate (eGFR). However, these parameters reflect the damage that has already occurred, limiting their utility for early detection. Therefore, novel biomarkers that can detect CKD before functional decline are urgently needed. Kidney fibrosis is an important pathologic feature of CKD and precedes functional impairment, highlighting its potential as an early indicator of CKD. Indoxyl sulfate (IS), a gut-derived uremic toxin, promotes kidney fibrosis and accelerates CKD progression, highlighting its strong potential as a biomarker for early CKD detection. As most studies on IS focused on the state of impaired renal function, the trend of IS in early-stage or preclinical CKD is unclear. Furthermore, quantifying IS in clinical practice is problematic because it's primarily measured through high-performance liquid chromatography. A recently developed enzyme method has simplified large-scale IS measurement. Identifying the associated factor with IS, we measured serum IS using enzyme method. We analyzed 674 participants who received specific health checkups and had an eGFR of ≥60 mL/min/1.73 m$^2$ (non-CKD) after obtaining their consent. In addition to standard examinations, we measured serum IS and administered a questionnaire about the frequency of defecating conditions. In this study, we investigated the level of the serum IS in association with the background characteristics of non-CKD participants. For the first time in this study, the median serum IS level was 3.7 (interquartile range, 2.7–5.0) μmol/L in individuals with non-CKD participants, and the linear regression analysis revealed that serum IS was significantly correlated with age, eGFR, and constipation. These findings suggest that IS accumulation

**Data availability statement:** All relevant data are within the manuscript and its Supporting Information files.

**Funding:** The author(s) received no specific funding for this work.

**Competing interests:** The authors have declared that no competing interests exist.

begins before overt renal dysfunction, supporting its potential as an early biomarker for CKD. Although further longitudinal studies are warranted, our results highlight the clinical relevance of IS in the early detection and intervention of CKD.

## Introduction

In 2017, the estimated prevalence of chronic kidney disease (CKD) was 9.1% worldwide, which was approximately 30% higher than in 1990. The estimated number of deaths was 120 million, with cardiovascular disease-related deaths exceeding 140 million [1]. In addition, a previous study using a simulation model reported that approximately 50% of non-CKD subjects may develop CKD [2]. Thus, it is anticipated that CKD will significantly impact global health. Estimated glomerular filtration rate (eGFR) and urinary albumin levels are used to evaluate and estimate renal function prognosis. However, these are insufficient for the early detection of CKD and selection of high-risk CKD cases because they indicate the result of kidney injury [3]. Therefore, it is necessary to develop new markers to detect CKD; currently, no biomarker exists for this condition [4,5]. Fibrosis is a pathological feature of CKD that leads to chronic organ damage [6]. Several studies have been conducted to assess potential novel biomarkers for kidney fibrosis; however, no biomarker is being used in clinical practice. Therefore, using a combination of biomarkers may be better than a single biomarker for renal function assessment [7]. Indoxyl sulfate (IS) leads to kidney fibrosis by activating mechanistic/mammalian targets of rapamycin complex one and nuclear factor-kappa B [8,9], and promotes kidney failure [10,11]. Reduction in IS accumulation suppresses kidney fibrosis [12,13]. Given this mechanism, it is anticipated that IS can be used as a biomarker for the early detection of CKD [14]. When disease-specific conditions damage the kidney, decreased glomerular filtration occurs due to the reduced number of functioning nephrons. A decrease in glomerular filtration leads to reduced IS excretion and higher IS levels in the blood. Higher IS causes excessive stress on the remaining nephrons, resulting in glomerulosclerosis and tubulointerstitial damage, which place additional strain on the remaining nephrons. The resulting decline in renal function leads to the further accumulation of IS, which in turn worsens the decline in renal function. Thus, a vicious cycle of progressive kidney damage is established, and IS promotes kidney failure [10]. However, the dynamics of IS in the kidney's preinjury (normal renal function) period remain unclear. The instantaneous measurement of IS and its application in clinical practice are difficult as IS is measured using high-performance liquid chromatography (HPLC) [15]. Furthermore, most previous studies on IS have focused on patients with impaired renal function; only a few studies have focused on patients with undamaged renal function [10,16–19]. An enzymatic assay kit has been recently developed as an effective approach to measure IS levels. An automated clinical chemistry analyzer can measure IS in 10 minutes. The IS values measured using this kit positively correlated with HPLC results (r = 0.933) [20]. This kit has also been validated for measuring various inhibitors in hemodialysis patients [21] and is expected to be used for the large-scale measurement of IS.

In this study, the enzyme method was used to measure IS on a large scale and to investigate IS and factors associated with IS in a general population with undamaged renal function, i.e., having an eGFR of ≥60 mL/min/1.73 m². 

## Materials and methods

### Study setting and participants

This study underwent from 01-06-2022 to 31-10-2022. The KDIGO guideline defines abnormal renal function as an eGFR of <60 mL/min/1.73 m² [22]. The main purpose of the present study was to assess serum IS in subjects with normal kidney function, defined as an eGFR of ≥60 mL/min/1.73 m², and to clarify factors affecting serum IS levels. In the participants in this study, normal renal function was defined as an eGFR of ≥60 mL/min/1.73 m². A questionnaire was administered to record the following data: age, sex, hypertension, diabetes, smoking history, and constipation. Moreover, the height, weight, body mass index, and systolic blood pressure of the participants were measured, and blood tests (HbA1c, creatinine [Cr], eGFR, and IS) and urine tests (albumin, Na, and Cr) were conducted. The Tanaka formula was used to estimate daily salt intake using spot urine samples [23]. Of note, the Tanaka formula was used in several research studies [23–26].

$$\text{Estimated daily salt intake (g/day)} = \left(21.98 \cdot \left(\frac{\text{Spot urine Na}}{\text{Spot urine Cr}} \cdot \frac{1}{10}\right) \cdot ((\text{Body weight} \cdot 14.89) \right.$$
$$\left. +(\text{Height} \cdot 16.14) - (\text{Age} \cdot 2.04) - 2244.45)\right) \cdot \frac{0.392}{17}$$

An enzyme-based assay determines serum Cr levels, whereas the Japanese estimated GFR formula calculates eGFR [27]. Of note, this formula was used in several research studies [28–30].

$$\text{eGFR in males (mL/min/1.73 m}^2\text{m2)} = 194 \times \text{Cr}^{-1.094} \times \text{Age}^{-0.287}$$

$$\text{eGFR in females (mL/min/1.73 m}^2\text{)} = 194 \times \text{Cr}^{-1.094} \times \text{Age}^{-0.287} \times 0.739$$

Serum IS was measured using an enzyme method [20]. The IS Assay Kit "NIPRO" (NIPRO Corporation, Osaka, Japan), which contains a reagent to measure total (free and albumin-bound) IS through the enzyme-based method, was used. The measurement principle of this reagent in the enzyme method is as follows: sulfatase converts IS to indoxyl. The generated indoxyl reacts with tetrazolium salt and produces the dye formazan. Formazan formation is measured by checking the absorbance at 450 nm, directly proportional to IS concentration. The total IS concentration is calculated by checking the absorbance at 450 nm of the standard IS solution that the manufacturer supplied. The enzyme method was developed to measure serum IS levels using a biochemical analyzer while measuring factors such as Cr. In this study, hypertension and diabetes were defined only by these measurements, and it was not considered whether the participants took medication for these conditions. In this study, hypertension was defined as a systolic blood pressure of more than 140 mmHg, the average of twice the systolic blood pressure measurement, and diabetes mellitus as more than 6.5% of HbA1c levels. Based on a previous report, a questionnaire was used to determine the frequency of defecation as follows: more than once a day, once every 2–3 days, once every 4–5 days, less than once every 6 days [31]. In the present study, we used this multiple-choice questionnaire, and constipation was defined as at most once every 2 days. The participants were categorized into the non-constipation group, defined as those with defecation more than once a day (≥1 time/day), or the constipation group, defined as those with defecation at most once every 2 days (≤1 time/2days).

The relationship between serum IS and other clinical factors was considered to determine the utility of measuring serum IS.

## Statistical analysis

Data on background characteristics influencing serum IS levels were presented using medians with interquartile ranges or percentages. The serum IS levels were not normally distributed; therefore, serum IS levels were logarithmically transformed to achieve a normal distribution. Other items weren't normal distribution too. These items were analyzed using Mann-Whitney U test, and Pearson $\chi^2$ test. Linear regression analysis was performed to evaluate the effect of covariates on serum IS. Linear regression analysis included nine covariates: age, eGFR, estimated salt intake, urinary albumin–creatinine ratio (U-Alb/Cr), sex, prevalence of hypertension, prevalence of diabetes, smoking history, and constipation. The regression model did not include Cr because of the collinearity between Cr and eGFR. Body mass index (BMI) was not included in the model, because BMI may differ from traditional health standards in elderly individuals whereas the present study included participants with a wide age range [32]. All statistical analyses were performed using EZR Ver1.64 (Saitama Medical Center, Jichi Medical University, Saitama, Japan). EZR (The R Foundation for Statistical Computing, Vienna, Austria) [33] is a statistical program that extends the functionality of R or R Commander. Statistical significance was defined as a P-value below 0.05.

## Ethical statement

This study was approved by the ethical committee of The Iwate Medical University School of Medicine (MH2021−181). This was an observational study, and participants who agreed to participate by signing written consent forms were enrolled.

## Results

### Serum IS concentration

Fig 1 shows the flow chart of participant enrollment. In total, 1102 subjects aged 40–74 years underwent a specific health checkup in Kanegasaki Town, Iwate Prefecture, in June 2022. Of these, 226 subjects did not consent to participate in the present study and 202 participants were excluded because of kidney dysfunction, based on an eGFR of <60 mL/min/1.73 m²). After excluding these participants, 674 participants were finally included in this study.

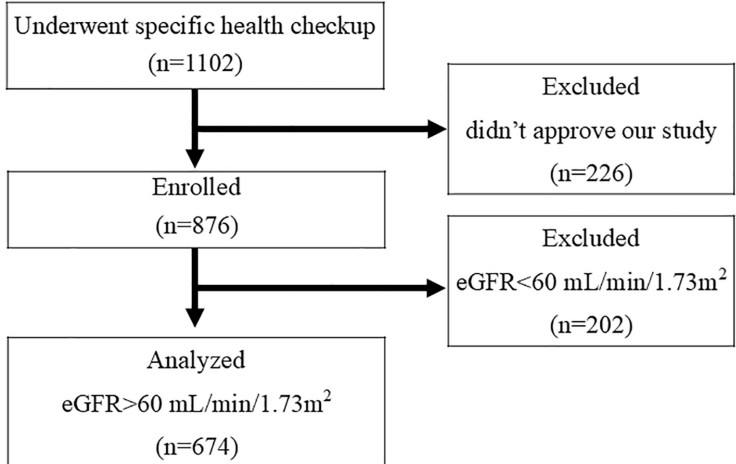

**Fig 1. Flowchart of selection.** A total of 1,102 subjects underwent a specific health checkup. Of these, 226 subjects who did not consent to participate in the study and 202 subjects with impaired renal function were excluded. Consequently, 674 participants were included in the study.

Table 1 summarizes the participants' characteristics. Male accounted for 47.2% (n = 318) of the total participants, with a median age of 68.0 (IQR: 63.0–71.0) years. The prevalence rates of hypertension and diabetes were 28.0 (n = 189) and 8.8% (n = 59), respectively.

Approximately 18.0% (n = 121) of the study participants smoked. The median serum Cr concentration, eGFR, U-Alb/Cr ratio, and estimated daily salt intake were 0.69 (IQR: 0.62–0.81) mg/dl, 72.5 (IQR: 67.2–79.7) mL/min/1.73 m², 10.6 (IQR: 6.1–19.1) mg/gCr, and 9.3 (IQR: 7.9–10.9) g/day, respectively. The prevalence of constipation was 21.8% (n = 147) of the participants.

Serum Cr, BMI, and smoking rate were all significantly higher in male than in female. Conversely, female had a significantly higher U-Alb/Cr ratio than male (Table 1).

The median IS level in the participants was 3.7 (IQR: 2.7–5.0) µmol/L, with males having an IS level of 3.9 (IQR: 2.6–5.4) µmol/L and females having an IS level of 3.7 (IQR: 2.8–4.8) µmol/L (Fig 2). The Q-Q plot of the serum IS levels indicated a non-normal distribution whereas the Q-Q plot of the logarithmically transformed IS levels indicated a normal distribution. There was no significant difference between males and females in terms of IS levels. There was also no difference in the prevalence of hypertension, prevalence of diabetes mellitus, and smoking history between males and females (Table 1). Age, Cr, eGFR, and prevalence of constipation were significant differences in characteristics of the participants between the groups with IS above the median and those with IS below the median (S1 Table). Significant differences were observed in serum IS levels between constipation group and non-constipation group. (S2 Table). No significant differences were observed in serum IS levels between groups with and without hypertension, diabetes mellitus, or smoking (S3–S5 Tables).

With serum IS concentration as the objective variable, linear regression analysis was performed with the following nine explanatory variables: age, eGFR, estimated salt intake, U-Alb/Cr, sex, prevalence of hypertension, prevalence of diabetes, smoking history, and prevalence of constipation. Age, constipation, and eGFR were significantly correlated with serum IS concentration. The serum IS value increased with age and constipation but decreased with eGFR (Table 2).

**Table 1. Characteristics of the participants.**

| | Male (n = 318) | | Female (n = 356) | | *p-value* |
|---|---|---|---|---|---|
| | Median | Quartile range | Median | Quartile range | |
| **Age (year)** | 68 | 62.0–71.0 | 67 | 63.0–71.0 | 0.597 |
| **Cr (mg/dL)** | 0.82 | 0.75–0.89 | 0.62 | 0.57–0.67 | **<0.001** |
| **eGFR (mL/min/1.73 m²)** | 72.9 | 67.1–79.9 | 72.4 | 67.3–79.6 | 0.812 |
| **BMI (kg/m²)** | 23.5 | 21.5–25.9 | 22.4 | 20.2–24.8 | **<0.001** |
| **Estimated volume of salt intake (g/day)** | 9.4 | 8.0–11.1 | 9.2 | 7.9–10.7 | 0.123 |
| **IS (µmol/L)** | 3.9 | 2.6–5.4 | 3.7 | 2.8–4.8 | 0.107 |
| **Urinary albumin–creatinine ratio (mg/gCr)** | 8.8 | 4.7–16.9 | 12.8 | 7.4–23.6 | **<0.001** |
| | Case | % | Case | % | |
| **Prevalence of hypertension n(%)** | 86 | 27.0 | 103 | 28.9 | 0.687 |
| **Prevalence of diabetes mellitus n(%)** | 31 | 9.7 | 28 | 7.9 | 0.415 |
| **History of smoking n(%)** | 99 | 31.1 | 22 | 3.2 | **<0.001** |
| **Prevalence of constipation n(%)** | 50 | 15.7 | 97 | 27.2 | **<0.001** |

The table shows the characteristics of all participants, males, and females. The median age of all participants was 68.0 years and serum IS level was 3.7 µmol/L. Serum Cr, BMI, smoking rate, and urine albumin–creatinine ratio significantly differed between males and females.

eGFR: estimated glomerular filtration rate, BMI: body mass index, IS: indoxyl sulfate

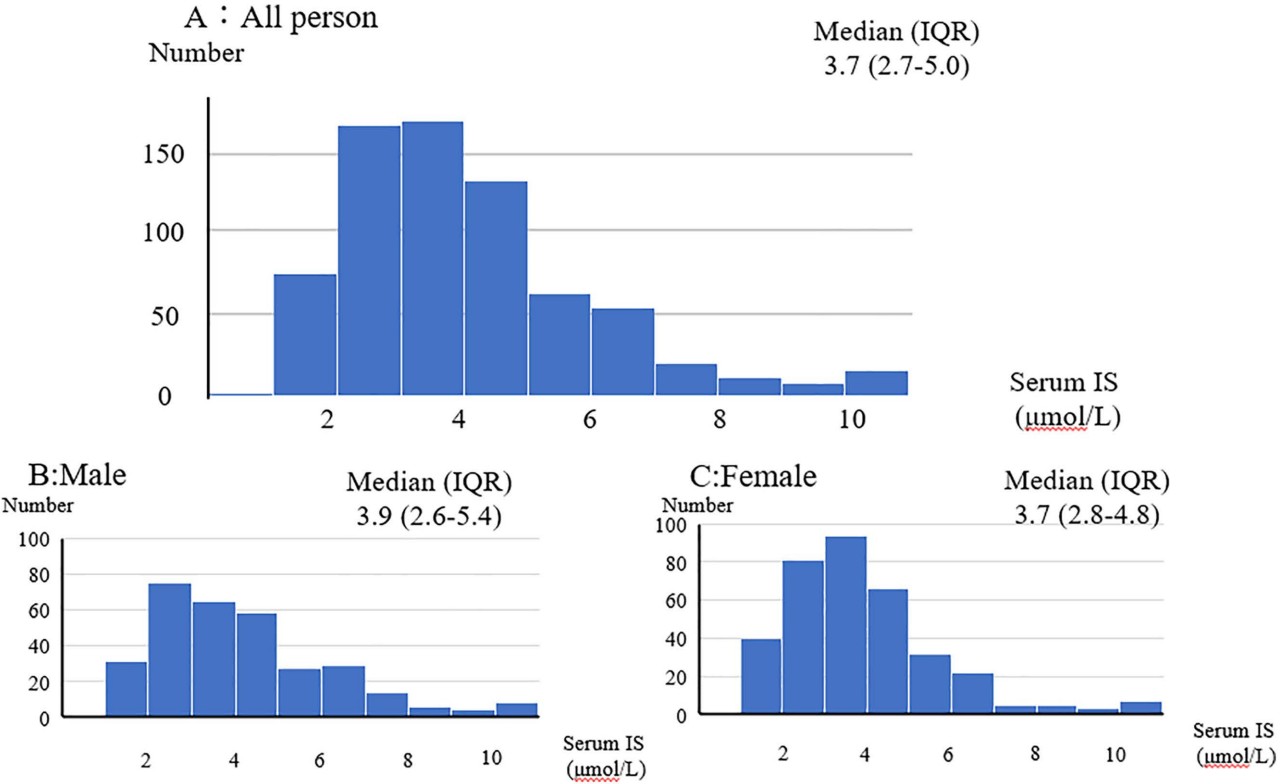

**Fig 2. Serum IS concentration.** The median IS level in all participants was 3.7 (IQR: 2.7–5.0) µmol/L, with males having an IS level of 3.9 (IQR: 2.6–5.4) µmol/L and females having an IS level of 3.7 (IQR: 2.8–4.8) µmol/L.

**Table 2. Linear regression analysis of IS vs nine items.**

|  | (n = 674) | | |
|---|---|---|---|
|  | **Regression coefficient estimates** | **95% confidence interval** | ***p*-value** |
| Age | 0.006 | 0.002, 0.011 | **0.008** |
| eGFR | −0.004 | −0.007, −0.001 | **0.022** |
| Estimated volume of salt intake | 0.005 | −0.011, 0.022 | 0.528 |
| Urinary albumin–creatinine ratio | 0.0002 | −0.0001, 0.0007 | 0.164 |
| Male sex | 0.076 | −0.001, 0.153 | 0.061 |
| Prevalence of hypertension | 0.014 | −0.067, 0.096 | 0.722 |
| Prevalence of diabetes mellitus | 0.082 | −0.046, 0.210 | 0.208 |
| History of smoking | −0.017 | −0.119, 0.085 | 0.744 |
| Prevalence of constipation | 0.162 | 0.075, 0.250 | **0.002** |

With serum IS concentration as the objective variable, linear regression analysis was performed on nine explanatory variables: age, eGFR, estimated salt intake, U-Alb/Cr, sex, prevalence of hypertension, prevalence of diabetes, smoking history, and prevalence of constipation.

eGFR: estimated glomerular filtration rate

## Discussion

The IS level of general population which their renal function was normal, defined as an eGFR of ≥60 mL/min/1.73 m² was 3.70 (IQR: 2.7–5.0) µmol/L (**Fig 2**). This level was the same as in previous reports [8,17–19]. In Japan, the prevalence rates of hypertension and diabetes were 30.6% [34] and 11.68% [35], respectively. Moreover, the rates of smoking history and estimated salt intake were 16.7% [36] and 9.95 ± 3.2 g [37], respectively. These were similar as our findings.

Previous studies reported that serum IS levels increased in parallel with the decline in renal function in patients with CKD. However, the dynamics of serum IS levels in individuals with normal renal function were unclear. In the present study, serum IS levels were measured in a large cohort, revealing that serum IS was also associated with eGFR in individuals with normal renal function, which was defined as an eGFR of ≥60 mL/min/1.73 m². Many biomarkers are thought to be useful in the early detection of CKD. However, these substances appear after an organ injury, for which there is no reliable marker [5,38]. Conversely, IS is a substance that causes kidney damage. The renal tubular cells' organic anion transporter (OAT1/3) and organic anion transporting polypeptide carry IS into the cell, thereby inducing oxidative stress and inflammatory cytokines through reactive oxygen species production [39,40]. IS also leads to the production of transforming growth factor-β by activating the renin–angiotensin aldosterone system, thereby causing kidney fibration [41]. Kidney fibrosis is an important biomarker for the early detection of CKD [6]. So, IS has the potential to act as an early detection marker. Furthermore, the gut microbiota outperforms the urine protein creatine ratio in detecting CKD [42]. This is because indole, a precursor of IS, is produced by the gut microbiota and may reflect changes in the gut microbiota. Although our study does not conclusively demonstrate the role of serum IS as an early biomarker of CKD, the longitudinal follow-up of the study participants long-term will evaluate this potential.

In this study, serum IS increased with age (**Table 2**), consistent with previous findings [18,19]. Aging-related changes in gut microbiota promote the conversion of tryptophan to indole [43,44].

Interestingly, constipation was also found to influence serum IS concentration (**Table 2**). Prolonged intestinal content transit time also increases the absorption of substances derived from the intestinal flora. Constipation affects the concentration of tryptophan metabolites in patients with CKD [45]. Moreover, the conversion rate to CKD and ESRD risk are higher in non-CKD patients with constipation than in those without constipation, and there is a correlation between the severity of constipation and the rate of eGFR decline [46]. Gut microflora produces several substances, with recent attention focused on the gut–cardio–renal axis [47], gut–immune–kidney axis [48], and brain–gut–kidney axis [49]. So, from the gut microbiota perspective, organ linkage should be considered using IS [50,51].

This study had some limitations: IS was measured as total IS rather than free IS, presence of potential bias in the racial population, protein intake from diet and gut microbiota was ignored, antimicrobial use was ignored, seasonal variability was ignored, eGFR was used to assess renal function, and this was a cross-sectional study that did not examine trends in renal function over time. Furthermore, many factors, such as PCS, contribute to kidney fibrosis (decreased renal function), and we did not consider these factors. Further research with many cases is warranted to examine the correlation of IS with precise renal function evaluation, investigate the relationship with gut microflora, and conduct prospective cohort studies on the impact of IS on renal function. In previous studies, the HPLC method was used to measure serum IS; however, measuring serum IS in real time is difficult because this method uses a dedicated machine with manual operation. On the other hand, the enzyme method can be performed in approximately 10 minutes on an automated biochemistry analyzer, making it easier to use in clinical practice. However, further comparison with conventional HPLC methods and clinical studies are needed. These aspects suggest that serum IS might detect CKD earlier than eGFR, which could not be evaluated in the present cross-sectional study and should be explored in future studies.

In the future, it will be necessary to predict the prognosis of CKD better and look for biomarkers that can help improve it.

## Conclusions

Serum IS was measured using an enzyme method in the general population with an eGFR of ≥60mL/min/1.73 m². The median serum IS level was 3.7 µmol/L in the general population. Age, eGFR, and constipation affected serum IS levels. Further research is needed to validate the potential of serum IS as a reliable biomarker.

## Supporting information

**S1 Table. Comparison of the participants with low and high IS levels. eGFR: estimated glomerular filtration rate, BMI: body mass index, IS: indoxyl sulfate.**
(PPTX)

**S2 Table. Comparison of the participants with and without constipation. eGFR: estimated glomerular filtration rate, BMI: body mass index, IS: indoxyl sulfate.**
(PPTX)

**S3 Table. Comparison of the participants with and without hypertension. eGFR: estimated glomerular filtration rate, BMI: body mass index, IS: indoxyl sulfate.**
(PPTX)

**S4 Table. Comparison of the participants with and without diabetes mellitus. eGFR: estimated glomerular filtration rate, BMI: body mass index, IS: indoxyl sulfate.**
(PPTX)

**S5 Table. Comparison of the participants with and without smoking history. eGFR: estimated glomerular filtration rate, BMI: body mass index, IS: indoxyl sulfate.**
(PPTX)

**S1 Data. Anonymized data files required to reproduce the analysis – original data.**
(CSV)

## Acknowledgments

We received technical advice from Kenta Tatsumi (Nipro Corporation).

## Author contributions

**Conceptualization:** Mizuki Hisano, Takaya Abe.

**Data curation:** Mizuki Hisano, Takaya Abe, Kie Sekiguchi, Eri Hakozaki, Toshikazu Abe, Kenichi Takeuchi.

**Formal analysis:** Mizuki Hisano, Takaya Abe, Kozo Tanno, Toru Maruyama, Ryo Takata.

**Investigation:** Mizuki Hisano.

**Methodology:** Mizuki Hisano, Takaya Abe, Fumiaki Takahashi, Kozo Tanno.

**Supervision:** Wataru Obara.

**Writing – original draft:** Mizuki Hisano.

**Writing – review & editing:** Takaya Abe, Toru Maruyama, Ryo Takata.

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
