## [Decision Letter · Decision Letter 0]

24 Sep 2025

Dear Dr. Hisano,

Thank you for submitting your manuscript to PLOS ONE. After careful consideration, we feel that it has merit but does not fully meet PLOS ONE’s publication criteria as it currently stands. Therefore, we invite you to submit a revised version of the manuscript that addresses the points raised during the review process.

We look forward to receiving your revised manuscript.

Kind regards,

Tatsuo Shimosawa, M.D., Ph.D.

Academic Editor

PLOS ONE

**Journal Requirements:**

1. When submitting your revision, we need you to address these additional requirements. Please ensure that your manuscript meets PLOS ONE's style requirements, including those for file naming. The PLOS ONE style templates can be found at https://journals.plos.org/plosone/s/file?id=wjVg/PLOSOne_formatting_sample_main_body.pdf and https://journals.plos.org/plosone/s/file?id=ba62/PLOSOne_formatting_sample_title_authors_affiliations.pdf 2. When completing the data availability statement of the submission form, you indicated that you will make your data available on acceptance. We strongly recommend all authors decide on a data sharing plan before acceptance, as the process can be lengthy and hold up publication timelines. Please note that, though access restrictions are acceptable now, your entire data will need to be made freely accessible if your manuscript is accepted for publication. This policy applies to all data except where public deposition would breach compliance with the protocol approved by your research ethics board. If you are unable to adhere to our open data policy, please kindly revise your statement to explain your reasoning and we will seek the editor's input on an exemption. Please be assured that, once you have provided your new statement, the assessment of your exemption will not hold up the peer review process. 3. If the reviewer comments include a recommendation to cite specific previously published works, please review and evaluate these publications to determine whether they are relevant and should be cited. There is no requirement to cite these works unless the editor has indicated otherwise. 

**Additional Editor Comments:**

Four experts raised several serious issues and some minor points. Please provide us revision and explain the rationale for statistical analysis and data variations.

Reviewers' comments:

**Comments to the Author**

1. Is the manuscript technically sound, and do the data support the conclusions?

Reviewer #1: Yes

Reviewer #2: No

Reviewer #3: Yes

Reviewer #4: Yes

2. Has the statistical analysis been performed appropriately and rigorously?

Reviewer #1: Yes

Reviewer #2: Yes

Reviewer #3: Yes

Reviewer #4: No

3. Have the authors made all data underlying the findings in their manuscript fully available?

Reviewer #1: Yes

Reviewer #2: Yes

Reviewer #3: No

Reviewer #4: Yes

4. Is the manuscript presented in an intelligible fashion and written in standard English?

Reviewer #1: Yes

Reviewer #2: Yes

Reviewer #3: Yes

Reviewer #4: Yes

**Reviewer #1: ** The authors have effected all the concerns raised by the reviewers. Therefore, the manuscript is technically sound and can be accepted for publication without further review. Also, their conclusion as:

Serum indoxyl sulfate (IS) levels were measured using an enzyme method in the general population with an estimated glomerular filtration rate (eGFR) of ≥60 mL/min/1.73 m². The median serum IS level in this population was found to be 3.7 μmol/L. Factors such as age, eGFR, and constipation were noted to influence serum IS levels. Further research is needed to establish the reliability of serum IS as a potential biomarker.

**Reviewer #2: ** This study investigates the association between indoxyl sulfate (IS) and kidney function in a non-CKD population, and as a study based on the gut–kidney axis, it is potentially interesting. However, the analysis was conducted in non-CKD participants, with no comparison to CKD patients, which limits the generalizability and interpretability of the findings. Furthermore, IS does not have a standardized clinical reference range, and therefore the clinical significance of variations within the presumed “normal” range remains unclear.

The study also lacks important analyses and visualizations that would strengthen the results, such as stratification by eGFR stage, scatter plots with regression lines and 95% confidence intervals, and correlation coefficients between IS and eGFR. As a result, the practical value and robustness of the data are limited.

While IS is generally expected to increase with declining kidney function, the conclusion that IS could serve as an early biomarker for CKD is not adequately supported in this cross-sectional design, as temporal precedence cannot be established.

Overall, the current data do not substantiate the abstract’s claim that IS may serve as a “potential early biomarker” for CKD.

**Reviewer #3:**  The manuscript reflects the authors' deep understanding of the field, and the conclusions are supported by the analysis results, with limitations discussed in detail. However, the authors will still need to address some concerns, including data availability, statistical analysis issues, and wording. Please refer to the attached comments for more details.

**Reviewer #4:**  I would like to thank you for the opportunity to review this manuscript. The study evaluates serum indoxyl sulfate levels in 674 non-CKD participants and its associations with baseline characteristics, discussing the potential of serum IS as an early biomarker for CKD. The manuscript is well structured, provides clear background and rationale for the study, and presents the results in an understandable manner. I have listed my comments below to assist in clarifying and improving the manuscript.

Major comments:

1. In the statistical analysis, the authors state using t-tests to compare group differences but reported median and IQR in Table 1. Please ensure consistency between the descriptive statistics and statistical tests. Typically, median (IQR) suggests a non-parametric distribution and would be paired with a Wilcoxon rank-sum test, whereas a t-test is generally reported with mean and SD.

2. In the statistical analysis, the linear regression models include covariates that may introduce multicollinearity (e.g., age already included in eGFR and daily salt intake formulas as well as a separate covariate). Please clarify whether this was assessed and consider reporting diagnostics such as Variance Inflation Factor (VIF) to verify multicollinearity does not bias the results.

3. In the results, the authors state that there are no differences in characteristics between IS groups in Table S1; however, age, Cr, eGFR, and constipation are statistically significant. Please revise the text to ensure consistency with the table.

4. Similar to my comment #3, in Table S2, IS is statistically significant between constipation groups, but reported no significant difference in the results section. Please revise for consistency between the table and text.

5. The study period (June–October 2022) was relatively short. It may be helpful to clarify whether this duration adequately captures seasonal variability and briefly discuss any potential influence of the COVID-19 period on the findings.

Minor comments:

6. In the abstract, “In this study, we investigated the level of the serum IS levels …”, the level is duplicated.

7. In second-to-last paragraph of the Introduction, the sentence with reference [15] uses the full name of HPLC but does not introduce the abbreviation. In the quoted sentence, “The IS values measured using this kit positively correlated with HPLC”, the abbreviation is directly used. Please add the abbreviation after the full term when first mentioned.

8. In the statistical analysis, the sentence “Linear regression analysis was performed to evaluate the impact of serum IS on outcomes” incorrectly suggests that serum IS is the independent variable, but serum IS is actually the outcome. Please revise the wording to clarify that serum IS was treated as the outcome.

9. In the results section, the sentence quoting Table S2-S5 refer to “comparison”; this should be “constipation”.

**Do you want your identity to be public for this peer review?** For information about this choice, including consent withdrawal, please see our Privacy Policy

Reviewer #1: **Yes: ** Musa Zakariah (PhD)

Reviewer #2: No

Reviewer #3: No

Reviewer #4: No

---

## [Author Response · Author response to Decision Letter 1]

13 Oct 2025

Reviewer #1: The authors have effected all the concerns raised by the reviewers. Therefore, the manuscript is technically sound and can be accepted for publication without further review. Also, their conclusion as:

Serum indoxyl sulfate (IS) levels were measured using an enzyme method in the general population with an estimated glomerular filtration rate (eGFR) of ≥60 mL/min/1.73 m². The median serum IS level in this population was found to be 3.7 μmol/L. Factors such as age, eGFR, and constipation were noted to influence serum IS levels. Further research is needed to establish the reliability of serum IS as a potential biomarker.

→I appreciate your comment. I will do further research to establish the reliability of serum IS as a potential biomarker.

Reviewer #2: This study investigates the association between indoxyl sulfate (IS) and kidney function in a non-CKD population, and as a study based on the gut–kidney axis, it is potentially interesting. However, the analysis was conducted in non-CKD participants, with no comparison to CKD patients, which limits the generalizability and interpretability of the findings. Furthermore, IS does not have a standardized clinical reference range, and therefore the clinical significance of variations within the presumed “normal” range remains unclear.

The study also lacks important analyses and visualizations that would strengthen the results, such as stratification by eGFR stage, scatter plots with regression lines and 95% confidence intervals, and correlation coefficients between IS and eGFR. As a result, the practical value and robustness of the data are limited.

While IS is generally expected to increase with declining kidney function, the conclusion that IS could serve as an early biomarker for CKD is not adequately supported in this cross-sectional design, as temporal precedence cannot be established.

Overall, the current data do not substantiate the abstract’s claim that IS may serve as a “potential early biomarker” for CKD.

→Thank you for your suggestion. Previous studies have demonstrated that IS increases with declining renal function. However, since no large-scale measurements had been conducted in individuals with normal renal function (eGFR ≥60 ml/min/1.73m2), we performed this study. As you pointed out, this study could not demonstrate fluctuations within the normal range or temporal precedence; further follow-up studies are necessary, as noted in the Limitations section. We will conduct follow-up studies in the future.

Reviewer #3

Statistical Analysis

1. One major issue is the data availability. It’s claimed by the authors that “All relevant data are within the manuscript and its Supporting Information files.” However, no original data or description regarding original data access is mentioned in the manuscript, except for the summary and model statistics based on the original data. PLOS ONE has a strict policy regarding data availability. It’s crucial for the authors to address this major issue accordingly.

→Thank you for your comments. We have uploaded the original data to Supporting Information files, but didn’t mention them in our manuscript. We have added that the original data can be accessed within its Supporting Information files. (Supporting Information p.23 L13)

2. It’s not rare to observe that low eGFR tends to correlate with high urinary albumin-creatinine ratio (UACR) in clinical studies regarding patients with CKD. Even though this study focuses on patients with no diagnosed CKD using the provided definition, it’s still recommended to add a little discussion to clear potential concerns over multicollinearity between eGFR and UACR, as they are both used in the linear regression model. Potential supporting evidences include the Pearson/Spearman Correlation Coefficient or the Variance Inflation Factor (VIF).

→Thank you for your opinion. The reviewer says that the relation between eGFR and UACR may have multicollinearity. We attach this question by calculating Pearson Correlation Coefficient. Correlation was 0.09, and 95%CI was from -0.01 to 0.13. Therefore, the relation between eGFR and UACR didn’t have multicollinearity.

3. “The regression model did not include Cr because of the collinearity between Cr and eGFR.” In the Statistical analysis section of Materials and Methods, it’s claimed that there exists a strong correlation between Cr and eGFR. However, it’s very strange that Cr is significantly different between the male and female groups in Table 1, while eGFR shows almost no difference between those two groups. Is the “eGFR” actually the “Urinary albumin–creatinine ratio”? Please double-check.

→Thank you for your comments. If linear regression analysis included ten covariates: age, eGFR, estimated salt intake, urinary albumin–creatinine ratio (U-Alb/Cr), sex, prevalence of hypertension, prevalence of diabetes, smoking history, constipation, and Cr, each VIF was as follows. Age 2.30, eGFR 11.94, estimated salt intake 1.09, urinary albumin–creatinine ratio (U-Alb/Cr) 1.03, sex 17.40, prevalence of hypertension 1.06, prevalence of diabetes 1.03, smoking history 1.47, constipation 1.03, Cr 27.88. Because the VIF of Cr was more than 5 and the highest VIF, we judged that Cr had a strong correlation between Cr and eGFR, and we didn’t add the covariates. (Statistical analysis section of Material and Methods p.8 L2-9)

4. It’s mentioned in the Statistical analysis section of Materials and Methods that the Student’s t-test was used to compare the continuous variables between the male and female groups. In this case, because means and variances are directly used in the Student’s t-test and p-values are also derived from it, it’s highly recommended to use means and 95% confidence intervals (CI) instead of medians and quartile ranges in Table 1. This also enhances the understanding of why some variables are significantly or marginally significantly different between the male and female groups, as the current statistics do not always seem to facilitate each other.

→Thank you for your comment. As the reviewer pointed out, all items are not normal distribution. So, we used Mann-Whitney U test and p-values were also derived from it. And Table 1 was used median and quartile range. (Result, p.10, Table1)

5. “20.2-74.8”

In Table 1, the third quartile of BMI for the female group is shown to be “74.8”. This number is very suspicious, given that the corresponding number from the male group is only “25.9” and there is no significantly higher percentage of diabetes mellitus observed in the female group. Please double-check this number.

→Thank you for your comments. The reviewer pointed out that the Table 1, the third quartile of BMI for the female group isn’t correct. The correct number is 24.8. I have revised it. (Result, p.10, Table1)

Wording

“Furthermore, adapting IS in clinical practice is problematic because it’s primarily measured through high-performance liquid chromatography.”

In the Abstract section, it’s a little unclear what “adapting” means in this sentence. If it means measuring, but “adapting” is used instead of “measuring” to avoid repeating because “measured” is used later, it’s recommended to use words like quantifying or evaluating. If it means adding IS evaluation to regular CKD clinical practice, it’s recommended to use words like incorporating or adopting. Any other words that make the statement clearer are also perfectly acceptable.

→Thank you for your comment. As the reviewer pointed out, “adapting” is unclear in this sentence. I have used “quantifying” in this sentence instead of “adapting”.(Abstract p.2, L11)

2. “Key Words: indoxyl sulfate, normal renal function, renal function markers” It’s more common to use “Keywords” instead of “Key Words”.

→Thank you for your comments. I have changed Key Words to Keywords. (Abstract p.3, L5)

3. “In 2017, the estimated prevalence of chronic kidney disease (CKD) was 9.1% worldwide, which was approximately 30% higher than in 1990. The estimated number of deaths is 120 million, with cardiovascular disease-related deaths exceeding 140 million [1].” In the Introduction section, the verb tenses are not consistent. In the first sentence, “was” is used, while “is” is used in the sentence after that. It’s highly recommended to re-examine the wording and keep the tenses consistent.

→Thank you for your suggestion. The verb tenses weren’t consistent. I have changed and keep the tenses consistent. (Introduction p.4, L3)

4. “When disease-specific conditions damage the kidney, decreased glomerular filtration due to the reduced number of functioning nephrons.” In the sentence from the Introduction section, a verb is missing after “decreased glomerular filtration”, such as “occurs”.

→Thank you for your suggestion. As you pointed out, a verb after “decreased glomerular filtration” was missing. I have added “occurs” after “decreased glomerular filtration”.(Introduction, p.4 L20)

5. “In the present study, constipation was defined as less than once every 2 days and the participants were categorized into the non-constipation group, defined as those with defecation more than once a day, or the constipation group, defined as those with defecation less than once every 2 days.” In the Study setting and participants section of Materials and Methods, two categories are mentioned regarding the constipation group categorization. However, it’s unclear how a category is assigned for subjects with average defecation frequencies of less than once a day but more than once every two days. Please consider making the definition clearer.

→Thank you for your comment. Our reference categorized into three defecation frequency groups: 1 times/day; 1 time/2-3 days; 1 time/4 days. Due to the limited number of subjects in our study who reported their defecation frequency, we categorized them into two groups. However, we confused the reviewer, so we revised our category accordingly. After that, there weren’t categorized subjects with average defecation frequencies of less than once a day but more than once every two days, and the definition was clearer (Material and Method, p.9)

6. In Table 1, it’s recommended to change the header “P” to “p-value” to be consistent with the format of Table 2.

→Thank you for your comment. The header in Table 1 has changed to “p-value”. (Results, p.9 Table 1)

7. In Table 1, the percentage format of the prevalence of hypertension for the male group should be consistent with other formats in the same table. Please keep one decimal place.

→Thank you for your comment. As the reviewer pointed out, the percentage format of the prevalence of hypertension for the male group in Table 1 has changed to “27.0”. (Results, p.9 Table 1)

8. In the footnote of Table 1, the first three lines might have a different font size from the last line. It’s recommended to double-check.

→Thank you for your comment. As the reviewer pointed out, the first three lines might have a different font size from the last line in the footnote of Table 1. I have changed the font size of the footnote in Table 1. (Results, p.10 Table 1)

9. In addition, it’s better to keep Table 1 and 2 formats consistent. For example, all cells in Table 1 are presented in bold fonts regardless of whether the p-values are significant or not, while only the significant p-values themselves in Table 2 are in bold fonts.

→Thank you for your suggestion. As the reviewer pointed out, the format of Table 1 and Table 2 wasn’t consistent. We revised the format of Table 1 only the significant p-values themselves are in bold fonts.　(Results, p.11 Table 2)

10. “The median IS level in the participants was 3.7 (IQR: 2.7–5.0) μmol/L, with males having an IS level of 3.85 (IQR: 2.6–5.4) μmol/L and females having an IS level of 3.7 (IQR: 2.8–4.8) μmol/L (Fig 2).” In the Serum IS concentration section of Results, “3.85” has one more decimal place than all other numbers reported in the same sentence. Please consider making it consistent with others, including changing it accordingly in Figure 2 and its legend. This issue is also observed elsewhere. Please revise all the number formats where applicable.

→Thank you for your comment. As the reviewer pointed out, “3.85” has one more decimal place than all other numbers reported in the same sentence. We revised Table 2 and Figure 2 in “3.9” to be consistent with others. (Results, p.10 Table 2, p.11 Fig 2)

Reviewer #4

Major comments:

In the statistical analysis, the authors state using t-tests to compare group differences but reported median and IQR in Table 1. Please ensure consistency between the descriptive statistics and statistical tests. Typically, median (IQR) suggests a non-parametric distribution and would be paired with a Wilcoxon rank-sum test, whereas a t-test is generally reported with mean and SD.

→Thank you for your comment. As the reviewer pointed out, all items are not normal distribution. So, we used Mann-Whitney U test and p-values were also derived from it. And Table 1 was used median and quartile range. (Result, p.10, Table1)

In the statistical analysis, the linear regression models include covariates that may introduce multicollinearity (e.g., age already included in eGFR and daily salt intake formulas as well as a separate covariate). Please clarify whether this was assessed and consider reporting diagnostics such as Variance Inflation Factor (VIF) to verify multicollinearity does not bias the results.

→Thank you for your comments. If linear regression analysis included ten covariates: age, eGFR, estimated salt intake, urinary albumin–creatinine ratio (U-Alb/Cr), sex, prevalence of hypertension, prevalence of diabetes, smoking history, constipation, and Cr, each VIF was as follows. Age 2.30, eGFR 11.94, estimated salt intake 1.09, urinary albumin–creatinine ratio (U-Alb/Cr) 1.03, sex 17.40, prevalence of hypertension 1.06, prevalence of diabetes 1.03, smoking history 1.47, constipation 1.03, Cr 27.88. Because the VIF of Cr was more than 5 and the highest VIF, we judged that Cr had a strong correlation between Cr and eGFR, and we didn’t add the covariates. (Statistical analysis section of Material and Methods p.8 L7-9)

In the results, the authors state that there are no differences in characteristics between IS groups in Table S1; however, age, Cr, eGFR, and constipation are statistically significant. Please revise the text to ensure consistency with the table.

→Thank you for your comment. As the reviewer pointed out, Age, Cr, eGFR, and constipation are statistically significant. We revised the text. (Result, p.10-11 L2)

Similar to my comment #3, in Table S2, IS is statistically significant between constipation groups, but reported no significant difference in the results section. Please revise for consistency between the table and text.

→Thank you for your comment. As the reviewer pointed out, Significant differences were observed in serum IS levels between constipation group and non-constipation group. We revised the text. (Result, p.11, L2-5)

The study period (June–October 2022) was relatively short. It may be helpful to clarify whether this duration adequately captures seasonal variability and briefly discuss any potential influence of the COVID-19 period on the findings.

→Thank you for your suggestion. As the reviewer pointed out, there seems to be seasonal variability since IS was the gut microbiota. But this study consisted of specific health checkup. So, we have added this sentence in limitation. (Discussion, p.14 L1.)

Minor comments:

In the abstract, “In this study, we investigated the level of the serum IS levels …”, the level is duplicated.

→Thank you for your comment. As the reviewer pointed out, the level is duplicated. We revised the text. (Abstract, p.2, L18)

7. In second-to-last paragraph of the Introduction, the sentence with reference [15] uses the full name of HPLC but does not introduce the abbreviation. In the quoted sentence, “The IS values measured using this kit positively correlated with HPLC”, the abbreviation is directly used. Please add the abbreviation after the full term when first mentioned.

→Thank you for your comment. We have added the abbreviation after the full term. (Results, p.5, L8)

In the statistical analysis,

---

## [Decision Letter · Decision Letter 1]

27 Oct 2025

Dear Dr. Hisano,

We look forward to receiving your revised manuscript.

Kind regards,

Tatsuo Shimosawa, M.D., Ph.D.

Academic Editor

PLOS ONE

Journal Requirements:

Reviewers' comments:

Reviewer's Responses to Questions

**Comments to the Author**

Reviewer #2: All comments have been addressed

Reviewer #3: (No Response)

Reviewer #4: All comments have been addressed

2. Is the manuscript technically sound, and do the data support the conclusions?

Reviewer #2: Yes

Reviewer #3: Yes

Reviewer #4: Yes

3. Has the statistical analysis been performed appropriately and rigorously?

Reviewer #2: Yes

Reviewer #3: Yes

Reviewer #4: Yes

4. Have the authors made all data underlying the findings in their manuscript fully available?

Reviewer #2: Yes

Reviewer #3: No

Reviewer #4: Yes

5. Is the manuscript presented in an intelligible fashion and written in standard English?

Reviewer #2: Yes

Reviewer #3: Yes

Reviewer #4: Yes

Reviewer #2: (No Response)

Reviewer #3: Thanks for the authors’ responses. Undoubtedly, the authors have reviewed all reviewer comments carefully and resolved most of my concerns. I just have a few follow-up comments based on my previous ones that would need the authors’ attention. The details can be found in the attached comments.

Reviewer #4: The authors have addressed all my previous comments and concerns. Thank you for the careful revisions.

**Do you want your identity to be public for this peer review?** For information about this choice, including consent withdrawal, please see our Privacy Policy

Reviewer #2: No

Reviewer #3: No

Reviewer #4: No

---

## [Author Response · Author response to Decision Letter 2]

5 Nov 2025

1. Data S1. Anonymized data files required to reproduce the analysis - original data. (CSV) Somehow, I could only access the “Suppl Table.pptx” file in the downloading link of Supporting Information. However, it was claimed that a CSV file with the original data was also uploaded, as written in both the manuscript and the authors’ response. It could be that the downloading link from my side might not have been updated. But it will be crucial for the authors to confirm that the original data can actually be downloaded using the downloading link in the manuscript.

→Thank you for your comment. I intended to upload Accessible Data to the Supporting Information section, but it didn't appear correctly. I sent an email to Editorial Office and confirmed the procedure. I uploaded it to the Supporting Information section as CSV file.

2. KeyWords: indoxyl sulfate, normal renal function, renal function markers Just like “Research” should never be written as “ReSearch”, it is inappropriate to use “KeyWords” instead of “Keywords”. Please revise it.

→Thank you for your comment. As the reviewer pointed out, it was inappropriate in this manuscript. I changed from “keyWords” to Keywords. (Abstract p.3, L5)

3. In the present study, constipation was defined as at most once every 2 days and the participants were categorized into the non-constipation group, defined as those with defecation more than once a day (≥1 time /day), or the constipation group, defined as those with defecation at most once every 2 days (≤1 time / 2days). I think the authors were confused about my previous comment on the constipation definition. I was thinking that the current definition was unclear in some scenarios. For example, if a subject had 4 defecations on different days of the past week, that would result in a defecation frequency of 4 / 7 = 0.57 per day. This value is neither ≥1 time/day nor ≤1 time/2days, and thus it’s confusing how this scenario should be classified. However, the data collection method might be specifically designed such that no confusion would be possible. For example, if subjects were asked how frequent their defecation was, with provided options like “daily”, “once every two days”, and “once more than two days”, the current definition would make sense. It will be helpful if the authors could bear in mind that there will be readers thinking like me in the way shown in the first paragraph, and try to avoid confusion in this case.

→Thank you for your comment. Our study used a multiple-choice questionnaire with options: more than once a day, once every 2–3 days, once every 4–5 days, and less than once every 6 days, with subjects choosing the answer. I revised the sentence to avoid misunderstanding. (Material and Methods, p.7, L13-14)

4. Figure 2 Somehow, there is a grey text box with Japanese characters in Figure 2. Please double-check if it was accidentally placed there or intentional.

→Thank you for your comment. As the reviewer pointed out, there was a grey text box with Japanese characters accidentally in Fig2. I have revised Fig2 in our manuscript. (Fig2)

---

## [Decision Letter · Decision Letter 2]

20 Nov 2025

The evaluation of indoxyl sulfate in the general population in Kanegasaki Iwate: A cross-sectional study (KANEGASAKI study)

PONE-D-25-26168R2

Dear Dr. Hisano,

We’re pleased to inform you that your manuscript has been judged scientifically suitable for publication and will be formally accepted for publication once it meets all outstanding technical requirements.

Kind regards,

Tatsuo Shimosawa, M.D., Ph.D.

Academic Editor

PLOS ONE

Additional Editor Comments (optional):

Reviewers' comments:

Reviewer's Responses to Questions

**Comments to the Author**

Reviewer #3: All comments have been addressed

2. Is the manuscript technically sound, and do the data support the conclusions?

Reviewer #3: Yes

3. Has the statistical analysis been performed appropriately and rigorously?

Reviewer #3: Yes

4. Have the authors made all data underlying the findings in their manuscript fully available?

Reviewer #3: Yes

5. Is the manuscript presented in an intelligible fashion and written in standard English?

Reviewer #3: Yes

Reviewer #3: (No Response)

**Do you want your identity to be public for this peer review?** For information about this choice, including consent withdrawal, please see our Privacy Policy

Reviewer #3: No

---

## [Editor Report · Acceptance letter]

PONE-D-25-26168R2

PLOS One

Dear Dr. Hisano,

I'm pleased to inform you that your manuscript has been deemed suitable for publication in PLOS One. Congratulations! Your manuscript is now being handed over to our production team.

Kind regards,

on behalf of

Prof. Tatsuo Shimosawa

Academic Editor

PLOS One